# Text, Method, or Goal? On What Really Matters in Biblical Thomism

Piotr Roszak 

Faculty of Theology, Nicolaus Copernicus University, Toruń 87-100, Poland; piotrroszak@umk.pl

**Abstract:** This article presents the history and main assumptions of biblical Thomism, which began with an attempt to restore interest in the biblical commentaries of Thomas Aquinas, but has managed to develop its own methodological procedure. The key to understanding it may be the idea of integration. Biblical Thomismism is thomistic in a mode that allows for, and encourages, direct engagement with theological and exegetical resources. Its aim is bringing together dogmatic, metaphysical, and exegetical modes into a contemporary theology that is Thomistic, ecumenical, and grounded in Scripture and the Fathers. This paper is divided into three parts. In the first part, the essence of the Biblical Thomism project is explained. Next, the general lines along which Biblical Thomism has been developing in recent times are detailed. Finally, an example of a proposed approach (*quies Dei*) is analyzed. In the conclusion, there is an outline of the emerging prospects for further research.

**Keywords:** biblical Thomism; medieval exegesis; hermeneutics; Church Fathers

## 1. Introduction

It might seem that the word "biblical" being placed next to the word "Thomism" would signify an escape into the biblical view of the world, thus validating the charge levelled by Whitehead nearly 100 years ago, when he diagnosed that the problem with theology was its persistent attachment to the old world-view (Whitehead 1967, p. 188). Classical theology, he maintained, was losing its ability to understand the surrounding world on a cultural level, its categories no longer matching the phenomena they were attempting to describe, with the inevitable consequence being that theology would become relegated to the backwaters of thought. According to that view, while science follows new categories, theology remains stuck in the "old" language. This entails a risk similar to that when a user fails to update a modern electronic device. If, for example, a mobile phone is not updated for a long time, it will eventually become impossible to send or receive messages. Whitehead's proposition is revolutionary: leave behind the old (the classical metaphysics of substance, biblical categories, etc.) and embrace the new (becoming).

In the same situation, St. Thomas would have responded differently: embrace both the old and the new, since the new does not develop by abandoning the old, just as the New Testament is not an ex nihilo creation that ignores the revelation of the Old Covenant, and by reading the Old Testament in even more depth, as perfectly illustrated by Thomas's commentary on Hebrews. This resembles the situation of the householder mentioned in the Gospel, who brings both old and new things from his treasury (cf. Matthew 13:52). Importantly, when he explains this short parable from Jesus in his commentary, St. Thomas interprets this image as the relationship between the Old Testament and the New Testament: the figure of Christ and adding new things to the Old Testament (Super Matt., c. 13). Thus, theology does not advance by abandoning and severing itself from the "old" or by inventing new constructs; instead, it delves deeper into pre-existing knowledge, in the spirit of the hermeneutic of continuity. This is the founding principle of Biblical Thomism: the language of theology should not discard the biblical vocabulary in favor of a modern one, but should instead foster the relationship with the source and update it within a new context. Hence,

Biblical Thomism is an art of integration that can be applied to all theology, rather than a mere aspect of the systematization of Aquinas's thought.

In view of the above, it is quite evident that Biblical Thomism began to establish a way to regain, first, the "old", that is the biblical commentaries that had been coming out of obscurity over the years, and then the tradition of the Church Fathers as the source of Aquinas's thinking, although taking into account what had taken place "after" Thomas (Vijgen 2016), among his commentators. The revived interest in St. Thomas's exegesis from C. Spicq and M.-D. Chenu OP has led, in the first place, to the realization that there is a biblical trait in his theology: a combination of speculative reasoning and interpretation of revelation. This is not about "extracting" further assertions from the biblical text but about combining what has been revealed—yet not always made evident in its interconnectedness—with systematic reflection (Boyle 2021, p. 134). Successive publications have pointed to historical issues concerning, not only the authorship of the commentaries, but also Thomas's sources, from the very text of the Bible that he used in his work (Bataillon OP), to his deference to the Church Fathers (Elders), to the philosophical citations appearing in the commentaries (Vijgen).

The impulse that contributed to the recognition of something more than a "renaissance" of commentaries was the research conducted by S. T. Pinckaers, recalling the biblical language that Thomas had neither abandoned nor treated as a burden. These new trends in approaching the biblical heritage are exemplified by J. P. Torrell OP, who has pointed out that biblical commentaries accounted for the majority of Aquinas's academic and monastic time, and are now essential in order to understand works of systematic theology. This movement of ideas is not unidirectional; instead, it acts as a feedback loop, whereby the reading of one enriches the reading of the other. Thus, Biblical Thomism was not born as a manifestation of theological "rebellion" or "guerrilla warfare" but as a consistently uncovered heritage. The years that followed have brought more in-depth analyses of the presence of Scripture in Aquinas's theological work, as well as of his procedures and their importance to the understanding of the roots of certain theological assertions, as expressed in W. Valkenberg's Words of the Living God (Valkenberg 2000).

In that vein, one can also approach M. Levering's seminal publication, *Scripture and Metaphysics* (Levering 2004), which addresses Aquinas's theological method and demonstrates that the Thomistic expression *ad Scripturam* (*ad fontes*) is about more than the text alone. As it has turned out, this is a confrontation of a style of theology that does not seek refuge in biblicism or treat Scripture as a loose collection of inspirations: taking note of "how" Thomas explains Scripture has the effect of revealing the participative importance of history in theology (Levering 2008). Revelation is not a linear set of events that resembles a kaleidoscope; instead, it is marked by God's involvement in each of these events. The deistic approach to revelation—whereby it is the text (rather than the hagiographer, through his prophetic charism (Rosemann 2015) that becomes inspired and therefore, as it were, trapped in the time of its creation—remains very far removed from Thomas's understanding of inspiration.

Thus, as we begin our reflection on what Biblical Thomism is, it is also important to note—as Aquinas would have done by beginning his reflection with *obiectiones*—what it is not (at least not exclusively): it is not a simple reminder of biblical commentaries, which account for approximately 13.5 percent of the entire corpus of Thomas's works, or a rejection of the Summa and other works as unimportant, nor is it a way of treating the Bible as a resource for quotations or arguments that could be taken out of context. Thomas had a scripturistic imagination that enabled him to discover connections between biblical texts, granting his theology a biblical format that harmoniously blended with philosophical advancements.

My presentation is divided into three parts. In the first part, I will explain the essence of the Biblical Thomism project. Next, I will discuss the general lines along which Biblical Thomism has been developing in recent times, providing a map, as it were, of this school

of thought. Finally, I will present an example of a proposed approach to thematic analysis that stems from Biblical Thomism and outline the emerging prospects for further research.

## 2. The Essence of Biblical Thomism: "Integration"

By analogy to music, employing a new method can always be compared to taking a new key that changes what one has previously been listening to. This is not a matter of altering the entire piece, the lyrics, or the score; instead, it is a matter of reading it in a different way than before and beginning to understand the interrelationships, implications, and dependencies. The result is a work in which there is no need to isolate individual bars and treat them as independent parts; instead, the focus is on discovering the mutual relationships. Biblical Thomism proposes a certain key that improves the clarity of the very theological project in which its exegesis plays an exemplifying role. The unique nature of this approach can be explained through several points, the key being the fourfold integration: doctrinal, theological, historical, and heuristic.

### 2.1. "Doctrinal" Integration: Tradition and Exegesis

What stands out as soon as one comes into contact with the biblical commentaries is the presence of philosophical tools evident in many exegetic procedures: questions, notes, or divisio textus. These tools were used to make it easier to see both the whole and the details, leading from reflection to analysis, that is breaking the text down into smaller units, which could then be synthesized in order to provide answers on the basis of the received Word.

Aquinas's biblical commentaries demonstrate a departure from monastic exegesis—which was based on the *lectio divina* and focused on the moral sense—towards a scientific and universalist exegesis that draws on Aristotle when it comes to explaining, for instance, grace, which requires understanding it as a *motus* or *necessitas*, or when it comes to explaining the necessity of Christ's cross for the salvation of man. There is no shortage of references to the Fathers, who are often quoted not only from the Glossa but also directly from the works that Thomas commissioned to be translated and happily included in his writings, such as the manuscripts of the Fathers' commentaries brought to him by Albert the Great; in fact, as noted by E. Alarcon, Thomas's work typically involved burrowing through the archives of the monasteries he visited (Alarcon 2012). Thomas juxtaposes the Fathers with one another, notices their different approaches and attempts to understand them—as in the case of the Antiochene dispute as to whether St. Peter's conduct was a venial sin or a mortal sin. In addition, there is the inclusion of dogmatic decrees of the councils (Morard 2005), passages from the Creed and philosophical reflections that require a rational approach. In short, a characteristic feature of Thomas's work is that he integrates the Tradition with a speculative approach instead of choosing one "against" the other. The aim of that endeavor is scripturistic contemplation, acceptance of revelation, and participation in the saints' knowledge of God (ST I, q. 1, a. 2c). This means that the emphasis in Biblical Thomism is methodological; it is an objection to the separation of theology from the Bible and, at the same time, to the reduction of theology to a mere repetition of biblical quotations. The origins of *sacra doctrina* lie in the acceptance and systematic, structured understanding of revelation.

### 2.2. "Theological" Integration: The Normative Value of Holy Scripture

In view of the above, the value of Thomas's exegetic procedures has to be interpreted from the perspective of his vision of theology. For him, biblical exegesis is, in a sense, an "alphabet" that makes *sacra doctrina* possible. It is built on the basis of the Word being read, which—like grace—does not operate "beside" or "above" nature, but for nature. One of the ways in which grace can operate, as emphasized by Thomas, is that is excites (*excita*) nature, stimulates it, and unleashes its power. A rational reading does not restrict the Word by making it conform to imperfect human structures. Instead, it explores the Word and discovers its associations: since theology is an "architectural" discipline, it builds a

synthesis, establishes relationships between the different components and, in doing so, discovers unity within a series of particular events. This process of discovery, in the light of the Word of God, is the most convincing when it takes place in a theology that is, at the same time, "a matter of argument". For Thomas, theology is a knowledge of "principles" rather than of "conclusions" that must simply be defended in a persistent manner (Pyda 2022).

For this reason, the presence of biblical texts in the *Summa theologiae* is not surprising, for they are included not only at the beginning of *sacra doctrina* but also throughout it: from lectio to disputatio to *praedicatio*. In fact, the last of these elements seems to demonstrate that the goal here is also *exitus–reditus*, coming from and returning to Scripture. This is not a springboard for exercises in speculative theology but an extended system of roots that produces fruit in the form of theological assertions. This means discerning in the biblical quotations their multiple roles: confirmative—when they offer a proof of an interpretation proposed by Aquinas; explicative—when they clarify the meaning of the text being commented upon; opening—when they open new theological questions; and deepening.

Biblical quotations also appear in the sed contra as part of the minor questions introduced in the commentaries. Here, they are used to address apparent contradictions between quotations (from biblical texts or from the Fathers) or to explain historical, as well as moral or doctrinal, discrepancies (Super 2 Cor., c. 1, l. 3, n. 26). In addition, they are frequently given at the end of the lectio, where they provide verbal concordance (making it clear that the reference is being made to the same word, although occurring in different contexts) or are linked by concurrent theological ideas (Roszak 2015).

Biblical Thomism reveals the biblical background of speculative theology at the level, not only of scriptural references, but also of theological concepts taken from the Bible, such as merit (*meritum*), which Thomas does not abandon and which he, instead, attempts to clarify. This fundamental biblicality is not negated by the presence of philosophical terms: in addition to the universalist dimension that the biblical message gains by referencing metaphysical concepts and to the academic nature of this exegesis, there is also a warning against the conceptual idolatry of biblical language, which is why metaphysics is needed when reading the Bible. In consequence, Biblical Thomism does not perceive metaphysics in exegesis as a "foreign body" that disturbs the pure waters of Scripture. The presence of philosophy (e.g., in the form of quotations from Aristotle in biblical commentaries, the terminology used, the philosophical problems mentioned, etc.) serves to demonstrate that the biblical message is open to everyone. This invalidates the charge levelled years ago by Pesch (Pesch 1974), who claimed that Thomas artificially made St. Paul a professor of theology. Thomas did not so much summarize Paul's works as he interpreted his Letters as being important, not only to their original recipients, but also to the successive generations of Christians. This means placing the Bible within the living practice of theology, showing how Scripture shapes *sacra doctrina* in its ceaseless effort to discover the truth about God.

In contrast to Peter Abelard's project, in which science was the starting point (Poirel 2016), Thomas's theology has its starting point in the Bible, and the confrontation with science stems from the fact that the supernatural truth can be known in its effects (including, for example, creation, ST II-II, q. 174, a. 5 co). In consequence, theology does not consist in exegesis alone but in the integration of exegesis with speculative theology. It is not surprising, therefore, that St. Thomas searches for a *propositum* in the commentaries, a doctrinal understanding of the text. For him, theology is biblical, narrational, and metaphorical, and the theologian's task is to express the biblical truth in a scientifically significant manner, so as to demonstrate its intelligibility (McGinn 2014, p. 65). Thus, *sacra doctrina* is a meeting point for the Bible and science.

This gives rise to an important feature, namely the alternativeness of exegesis—a feature characteristic of all of Thomas's theology, and which often presents different interpretations instead of providing the one and only correct answer. A passage from the Bible can be explained in a number of ways, and Thomas does not always point to the one that is *melior*, which acts as a safety device that prevents a fundamentalist reading of the Bible. As

a result, theology becomes a process of discovering the richness of the Word, an elucidation (Quodlibet VII, q. 7, a. 2c), as evident in the manner in which Thomas practices spiritual exegesis and in the value that he attaches to it.

### 2.3. "Historical" Integration: The Material and Formal Presence of the Fathers

The integration that characterizes Biblical Thomism includes recognizing the role of the Church Fathers in Aquinas's exegesis. Despite the conviction that a "jump" to the first century is necessary (as suggested by Barth) in order to gain access to revelation, it being thus understood quasi-deistically, St. Thomas accepts the *auctoritas* of the Fathers as partakers in the transmission of the Tradition. Taking their views into consideration and entering into a dialogue with them, Thomas incorporates them into the authority of the Church by pointing to the ecclesial context of biblical exegesis (as the correct hermeneutical horizon). Drawing from the texts of the Fathers is a manifestation of a certain theological continuity, to which Aquinas will remain faithful until the end, and works such as the *Catena aurea* are yet to be fully discovered and—even more importantly—understood in depth, so as to reveal "how" Thomas worked with these texts and for what purpose. Thomas does not consider the Church Fathers to be a separate source in relation to Scripture; instead, he believes that their works make possible a correct understanding of the biblical text (Roszak and Vijgen 2021, p. 9). This stems from the presence of the same Spirit who fills the hagiographers and the Fathers, acting upon both intellect and will, although the inspiration is obviously different in the two cases.

The above presence may, in some cases, have a material expression in that it manifests itself in specific quotations; in other cases, it has a formal dimension, when Thomas adopts the Fathers' manner of pursuing theology. It is evident that in his exegetical work, he attempts to juxtapose his own exegesis with that of the Fathers and with the truth about the world; this is not concordism but a patient and consistent effort to build a synthesis.

### 2.4. "Heuristic" Integration: The Literal Sense and the Spiritual Sense

Biblical Thomism revisits Thomas's theory of the biblical senses, a concept which theologians have begun to grasp more thoroughly in recent years. The primacy of the literal sense emphasized by Aquinas, inherited from the Victorines, does not mean eliminating other senses or simply preferring one manner of interpretation. Instead, it means applying a more methodical procedure, which is something that came to the foreground in the famous dispute between Henri de Lubac (1998) and Beryl Smalley (1952). Biblical Thomism seeks to demonstrate that the literal sense is a starting point, upon which the spiritual sense can subsequently be developed. Thus, a theologian is not faced with two parallel paths, between which he or she can choose by following either the literal sense or the spiritual sense. In other words, the former is not a goal in itself but a step on the way to the latter. The difference in importance between the two senses in exegesis reflects the fact that the literal sense plays an argumentative role in theology, but that does not diminish the value of the spiritual senses. A spiritual interpretation of the New Testament is its literal sense (Manresa 2017).

Perhaps here, too, there is an opportunity to apply the theory of hylomorphism, according to which the literal sense (matter) and the spiritual senses (form) together determine the overall sense. This integrative intuition manifests itself in a concern for the literal sense, which to mediaeval biblical scholars did not mean separating all the tiny particles of allegorical readings using a scalpel and tweezers. Such an approach leads to the search for *consensus* rather than difference.

This procedure can be exemplified by the manner in which the words of one of the Psalms are interpreted: "He gathered the waters of the sea as in a bottle" (Psalm 33:7). Aquinas explains them in the literal sense as a reference to the order of the world in which water (as in a vessel) does not flow out but is contained; it is drawn for use and does not vanish. In biblical language, it is a prerogative of God the Creator to contain the sea. In that context, Aquinas derives the etymology of the word *abyssus* from *a–bassis*, meaning

"without foundation". In the spiritual sense, he demonstrates that the vessel may, on the one hand, represent good men: peoples come together in the Church, as in a wineskin, a container made from the skin of a dead animal, and thus mortify themselves. Another interpretation points to converted sinners who had previously lived in the abyss of vices (Paul, Matthew, Magdalene). The depths or abysses can also be interpreted as biblical senses that are deposited in the storehouses of the Sacred Scriptures. On the other hand, if the image is interpreted as pertaining to evil men, then it means suffering and agony on the outside and mercy on the inside. In the anagogical sense, this is a sign that the persecutors of the Church will be gathered in an abyss over which God stands watch (In Psalm. 32, n. 311).

### 3. Trends in the Development of Biblical Thomism

When attempting to draw a "map" of Biblical Thomism, it is worth noting that the development of this school of thought has been driven, so to speak, by three main objectives. First, to gain ever greater knowledge of the textual content, chronology, and theological value of biblical commentaries, as well as specific biblical quotations that appear in different contexts. Second, to identify the purpose of this theological practice, which is to gain sapiential knowledge capable of interpreting reality in the light of the most fundamental reasons. Third, to practice a method of analyzing theological subjects that are based on the exemplaristic paradigm.

### 3.1. Biblical Texts in Aquinas's Theological Practice

Without doubt, Biblical Thomism can be credited with restoring the value of biblical commentaries, which had for centuries been overshadowed by works in systematic theology. While their existence had not been completely forgotten, the value of Scripture to the idea of theology itself had clearly been disregarded. What I am referring to here is not a simple commentary on the biblical text that would constitute a goal in itself, but rather the beginning of a theological journey: the lectio was biblical and, through the disputatio, it endeavored to bring everything together in the *praedicatio*. In Thomas's work, the biblical text is at the center, with everything else gravitating around it: doctrinal syntheses feed upon references to scriptural texts and respect their guidance. This demonstrates that Thomas possessed a scripturistic imagination. Raised, as it were, on the Holy Bible, from Monte Cassino to Naples, he constantly related dogmatic truths and philosophical arguments to specific biblical texts (SCG III, c. 64 § 9). This is reflected in the practice of his exegesis, in which he would—often in a manner surprising to the modern reader—relate the truths of faith to the events described in the Bible (Super 1 Cor., c. 9, l. 4, n. 496). In view of all these points, one might ask whether the *Summa theologiae* was written "for" the commentaries or vice versa. While there certainly is some feedback, it is also clear that some of Aquinas's works, such as the Summa, cannot be taken in isolation from the Bible. To treat them as "self-contained" pieces without any biblical *utilitas* would be inexplicable from the standpoint of Aquinas's concept of *sacra doctrina*.

The modern discovery of the biblicality of the *Summa theologiae* is taking place on several levels. There are initiatives that focus on the identification of biblical material in the *Summa*. At the same time, however, there are also attempts to establish how specific biblical texts function throughout the Summa and in which topics they appear as arguments, e.g., Romans 1:19–20 (Ebert 2020), or which part of a given quotation is invoked in a given context, because it is evident—as with 2 Peter 1:4 (*consors divinae naturae*)—that Aquinas sometimes quotes the first part and sometimes the second part of a phrase (Spezzano 2015).

Another approach is to examine a particular passage within the corpus of the works of St. Paul, which is what W. M. Wright did when he analyzed Thomas's interpretation of Galatians 3:28 in the light of MacIntyre's hermeneutic assumptions concerning the role of tradition (and thus the rationality of community). Similarly, Shawn Colberg discovered biblical documentation in Aquinas's deliberations on reward and grace, demonstrating that many of Thomas's theses were in fact based on Scripture (Colberg 2020). Worthy of note are also the attempts to interpret the same biblical texts depending on how they are used in

the *lectiones*, then in an argumentative role in the *disputatio*, and finally in the *praedicatio*. The manner in which Thomas would resolve the apparent contradictions between biblical quotations was addressed by M. Przanowski OP, who focused on two quotations: Christ being described as "taking the form of a servant" (Philippians 2:7) and, at the same time, being referred to in the Gospel of John as "full of grace and truth" (John 1:14) (Przanowski 2018). By way of juxtaposition, it is possible to discover certain theological preferences in Aquinas, of which a reader of the commentaries should be aware.

It is also important to consider the manner in which biblical texts appear in other commentaries, such as those concerning the works of Pseudo-Dionysius, Boetius, or Aristotle, in which the classic expression *consonat Scriptura* appears—as is the case when Aristotle's *De anima* is juxtaposed with the biblical account of the soul and its ability to know the truth (Super De Trinitate I, q. 1, a. 1c).

What is the point, however, of using direct biblical quotations in theological argumentation? The answer is as follows: to combine the two aspects in order to obtain a complete picture, instead of juggling with quotations taken out of context. Thomas strives for a comprehensive approach: a synthesis of one and the other. Sometimes, this requires an even deeper intervention, since what matters is not only the overtone but also the orientation towards specific texts in which a given reading requires interpretation. In some cases, Thomas observes that a biblical phrase can be divided in different ways, depending on whether the teleological meaning or causative meaning is being studied, but the motivation behind his meticulous divisions is to reflect upon the different senses of a biblical text.

The biblical text is, therefore, being considered within a certain scheme that recurs and retains a similar structure: the division of the text, the notae that explain broader contexts, the intriguing quaestiones, and the references to the etymology of words and sometimes local sayings. Aquinas's biblical commentaries are not a theologian's free spiritual reflections; instead, they demonstrate a structured approach to the text that stems from the scholastic method. The above formalization of exegesis is undoubtedly a product or effect of a scientific understanding of theology.

### 3.2. Scripturistic Contemplation

Biblical Thomism helps highlight the sapiential character of theology, which is a wisdom-oriented knowledge (since it concerns God as the ultimate goal) that enables one to partake in grateful contemplation (Case 2016). Being wise means being able to interpret the world in the light of the most fundamental reasons. This aspect is particularly evident in the *praedicatio*, which typically contains a question about the purpose, about why something is undertaken. Here, the answer that follows is an attempt to combine many possible interpretations and thus arrive at the most appropriate one.

The goal of Biblical Thomism is to encourage contemplation of the truth that is attested to in the biblical text and to draw attention to the purpose for which the Bible was written in the first place. This question is particularly relevant in the time of the "hermeneutical fog" which has emerged with historical-critical exegesis and can be dispersed by reclaiming the above purpose.

What matters in this form of contemplation, in grasping the truth, are the interpretations of the Church Fathers, linguistic analysis, and hermeneutics (built, for example, upon the division of the text: *divisio maior* and *minor*). With such a reading of Scripture, faith is the starting point, and the deepening and development of that faith is one of the goals. The key issue, however, is to orient the exegesis towards the truth, rather than towards emotional strengthening.

Themes that combine doctrinal (speculative) matters and Christian life (e.g., new creation) are an important part of Biblical Thomism. On the one hand, this gives rise to a number of issues concerning eschatology and the importance of "old" creation in relation to "new" creation (that is whether such creation will occur *ex vetere* or *ex nihilo*, or whether or not something will survive and pass on to eternity (Roszak 2022b); in fact, Thomas relates Jesus's remark that the hairs of our heads are all numbered in his commentary on

Matthew to the very question of the relationship between worldliness and eternity). This is why the exegesis points towards important truths of the faith, leaving the reader to relate them to his or her own life. What emerges from this method of working with the biblical text is a unique kind of Thomism: not a copy-and-paste Thomism of ready-made answers, but a Thomism that patiently builds the context, establishes a sense of direction, and constantly shows everyone where they are, thus addressing the meaning of human life within a broader vision. This is the nature of the *Summa theologiae*: it is not a collection of elements that somehow fit together but a discovery of the ordo of human life, of where one has come from and where one is going, of the meaning of everything.

*3.3. The Method of Biblical Thomism*

In recent years, the proposition of Biblical Thomism has been transformed, in practical terms, into a method of interpreting Aquinas's thought by applying an ordo-based formula that respects the importance of Scripture in his theology. The above method is not limited to merely analyzing biblical commentaries; instead, it incorporates speculation and relates Scripture to Christian life, which in the mediaeval framework often took place in the *praedicatio*. This methodological proposition takes into account Thomas's characteristic way of thinking in terms of exemplar–exemplum, placing God and His revelation—as attested to in Scripture—at the center and then contemplating this mystery to discover a number of references to the *modi* of a Christian's presence in the world.

The application of this Biblical Thomistic procedure can be demonstrated by an analysis of God's rest (*quies Dei*) after creation, as described in Genesis 2:2 (Roszak 2022a). Thomas discusses it in the context of his reflection on bodies at rest in the physical world and then relates it to the quiet life of a Christian (cf. 1 Thessalonians 4:11). The Latin word *quies* invokes, in the first place, a lack of motion (*privatio motus*), that is cessation of activity and attainment of a stable existence in a given place. However, there is also the rest of desire (*quies desiderii*) that comes with the achievement of, and "repose" in, the desired end (Enrique and Montoya 2021). In this sense, Thomas says that the will delights in the sought end (which he describes using the term *delectatio*), and contemplation leads to a rest in truth (which is what a "rest of conscience" would consist in), but it is the earthly experience of such fulfilment that acts as the inception (*inchoatio*) of an eternal rest in God. Hence, it is not surprising that this is what a Christian prays for in reference to the departed: *requiem aeternam*.

From this perspective, Thomas differentiates between two kinds of rest that help reach the truth about God's rest after creation:

> It should be noted with Augustine that he does not say simply that he rested, but that he rested from his works. For he rested in himself from all eternity, but when he rested, it was not in his works, but from his works. For God works in a different manner from other artisans; for an artisan acts because of a need, as a house builder makes a house to rest in it, and a cutlerer a knife for gain; hence, the desire of every artisan comes to rest in his work. But not so with God, because he does not act out of need but to communicate his goodness; hence, he does not rest in his work, but from producing a work; and he rests only in his goodness. (Super Heb. [rep. vulgata], c. 4, l. 1, n. 204)

In the case in question, therefore, this is not the end of a process (because creation would return to nothingness without the *creatio continua*) but an activity of resting in the goodness that is being communicated to creation. God's rest consists in Him knowing Himself, an act of contemplation through which God knows the world. God does not know things as external to Himself but in or through His essence and is happy by delighting in Himself in that manner. The achievement of *consummatio* means ceasing to create new beings and attaining two kinds of perfection: (1) by virtue of consisting of all the essential parts (*ex omnibus suis partibus essentialibus*), and (2) by virtue of the world being subordinated to its end (*ex ordine ad finem*).

God's productive rest is a model for human action, as Thomas notes in his commentary on Hebrews: "Just as in the old law the sabbath represented God's rest from his works (Gen 2:2), so too that rest will be that of the saints from their labours. From henceforth now, says the spirit, that they may rest from their labours (Rev 14:13)." (Super Heb., c. 4, l. 2, n. 209). This is not a case of inactivity, since the Redeemed are indeed active in the beatific vision.

Besides offering a literal explanation of rest, it is possible to identify its allegorical sense, whereby it signifies Christ's rest in the tomb, and its anagogical sense, whereby it signifies the soul's rest in God (Super Col., c. 2, l. 4, n. 120). This translates to the spiritual path of a disciple of Jesus: through baptism, the believer moves from Christ's rest in the tomb to being buried together with Him, and thus to partaking in His rest (Super Sent. III, d. 37, a. 5, qc. 3 ad 1).

A call to live "quietly" but not idly appears in 1 Thessalonians 4:11: *ut quieti sitis*. This refers to living a quiet life that is free from curiosity (as suggested by the quotation from Proverbs 7:11) but also to protecting Christians from restlessness (*inquietudo*), for the latter causes a great deal of damage by focusing a person's attention on secondary things—a consequence of the loss of original justice. This return to quietude comes largely from *continentia*, which restrains one's behavior and introduces order into the sphere of sensual impulses.

The journey from the discovery of the meaning of rest in the world of material beings, to the rest of desires and contemplation of sentient beings, to God's rest after creation has its important liturgical dimension, associated with the worship of God and the meaning of Sunday. For Biblical Thomism, this liturgical aspect and the search for it (as undertaken by Aquinas himself) are equally important. The procedure is, in essence, an attempt to apply the integrating method of *sacra doctrina* to theological matters and demonstrate how dogma determines conduct.

## 4. Conclusions

Since St. Thomas describes even the effects of grace in terms of "motion", one could ask about the direction in which Biblical Thomism is moving. I believe that Biblical Thomism strives to "move" the Thomistic view, so that, rather than looking at Thomas himself, we should look at what he observed and contemplated: the Scripture that bears witness to revelation. In this way, we should become convinced that behind the philosophical and theological constructs we have so admired, the starting point of all reflection is the Bible (Waldstein 1994). By grounding theological reflection in Scripture and incorporating various *auctoritates* in it, so as to integrate the message, and by founding such reflection on God's exemplarism at the same time, it is possible to demonstrate that theology is a knowledge of principles that, as a source of truth, enables one to gain a better understanding of one's circumstances.

What kind of theology does Biblical Thomism build? The answer is integrated but not integristic. It is a theological culture that relies on arguments and convincing ideas, abandoning the deistic understanding of revelation that reduces it to past events. The latter approach—similar to ignoring Scripture in theology altogether—only uses scriptural texts sparingly, as a mere confirmation of certain theses, or adds them as an embellishment rather than a pivot of thought, which does not foster the cultivation of *sacra doctrina* (Vijgen 2018).

In view of the above, Biblical Thomism stands in opposition to the narrow-minded view that only brilliant reasoning and argumentation matter, and that the Bible and biblical quotations serve a mere decorative purpose. The objective of Biblical Thomism is to understand what theology stands for as a field of knowledge that offers the key, listens, and provides structure. It should be noted that in his interpretation of God's response to the debate between Job's friends, Aquinas pays attention to the style in which God—making the *determinatio magistralis*—introduces His response (Job 39–42). The response has the form of questions that prompt one to search, presenting an overview of the created world and the (inter)relationships that exist in it: an animal park tour, so to speak, that enables

man to look at the world anew through a different lens. This shows that reading biblical commentaries does not mean isolating different aspects of Thomas's theological activity. On the contrary, it means understanding his proposition of *sacra doctrina* as a combination of speculation and revelation.

Biblical Thomism is fruitful in part because it can be in dialogue with other theological approaches grounded in Scripture, including with contemporary theology that is biblically rich (for instance, some instances of Ressourcement theology, such as Ratzinger's or Balthasar's), with Christian and Jewish "biblical theologies", and with the insights of historical-critical biblical scholarship, insofar as these insights interface with dogmatic theology. Biblical Thomism is Thomism, but in a mode that allows for and encourages direct engagement with the above theological and exegetical resources, with the aim of bringing together dogmatic, metaphysical, and exegetical modes into a contemporary theology that is Thomistic, ecumenical, and grounded in Scripture and the Fathers.

With regard to the future directions in which Biblical Thomism may develop, we can identify at least three areas that show promise:

(1)   reconstruction of commentaries which were not written by Thomas on the basis of quota-tions which can be found in systematic works or other commentaries. This method can be used to interpret, for example, the Song of Songs (Bonino 2019). A similar approach can also be used for the sapiential books or the Book of Genesis;

(2)   observation of how biblical quotations function in the different systematic works or in the commentaries on Dionysius, Boethius and Aristotle, thus explaining in more detail the normative character of Scripture for philosophical studies; and

(3)   increased interest in the history of biblical commentaries in the Thomistic school and in the reception and continuation of Aquinas's method: in this context, publication of Cajetan's biblical commentaries is a promising sign (O'Connor 2017).

In response to the question posed in the title, that is "text, method, or goal", one must answer as Aquinas would: *et–et* (one and the other) rather than *aut–aut* (either one or the other). Biblical Thomism suggests paying attention not only to the texts of biblical commentaries and to the theological method, but also to the purpose of the reflection being undertaken—a reflection which draws light for a Christian existence from truth about God. The goal is not to fortify and enclose theology in its language but to remind us that exegesis is an encounter with the living God (Wright IV and Martin 2019), and thus to open it to new themes, so that Aquinas's key can be used to unlock further challenges that face the wisdom coming from above—from the Father of Lights (cf. James 1:17).

**Funding:** This research was funded by National Science Centre in Poland, grant number 2019/35/B/HS1/00305.

**Acknowledgments:** My gratitude to Matthew Levering and Jörgen Vijgen for their fruitful comments and participants of the Eleventh International Thomistic Congress in Rome (19–24 September 2022).

**Conflicts of Interest:** The author declares no conflict of interest.

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
