# Peer review of "Text, Method, or Goal? On What Really Matters in Biblical Thomism"

_religions, doi:10.3390/rel14010003_

Round 1
Reviewer 1 Report
I believe that the Biblical Thomism project is a welcome endeavor because it shows that the interpretation of Holy Scripture is possible beyond the hegemony of historical-critical exegesis. As the author/authors of this article observe, Biblical Thomism reveals that biblical interpretation is a scientific, sapiential, and speculative enterprise. Thus, Biblical Thomism contributes to a better knowledge of the theology of Thomas Aquinas, and at the same time highlights hermeneutical principles in the interpretation of the Bible.
To what extent can Biblical Thomism represent an autonomous method of biblical exegesis? In my opinion, the answer to this question is negative, for a method presupposes a series of principles that can be applied to a biblical text, whatever it may be, for the purpose of understanding its message. On the other hand, the hermeneutic principles that Biblical Thomism identifies may be integrated or combined with other approaches/methods of biblical exegesis.
As for the format of this article, it is quite difficult to read because it describes condensed a project that would be presented much better as a book. Examples of biblical interpretation are given almost exclusively in section 3.3. Perhaps a restructuring of the manuscript, so that each of the theoretical statements may be accompanied by examples of biblical exegesis from the writings of St. Thomas Aquinas would be welcome. This could also be achieved by restructuring this manuscript as a series of two or more articles.
Author Response
Thank you very much for your comments. You are totally right that this should be develop further in more articles or even a book. To be honest, I am working on such a project. This is just a general overview of Biblical Thomism that is required, as far as I could note in many conferences, by various scholars from this field.
Reviewer 2 Report
This is an excellent essay. Occasionally, the English needs some attention; it seems likely that this is by a scholar whose first language is not English, although maybe I have imagined this. The concluding three points at the end of the essay for 'promising directions in which Biblical Thomism may go' seem weak to me. If these are the promising directions, then Biblical Thomism is not going to go far. The argument shortly prior to this that Biblical Thomism invites a dialogue with Ressourcement theology, with Jewish theology, with ecumenical biblical theologies was much more promising. My suggestion is either to improve the concluding three points or else to drop them. It is true that Bonino's work is valuable, and so that should be mentioned somewhere in the paper but maybe not in the conclusion.
Author Response
Thank you for inspiring comments. I have mentioned in the main part of my paper that Biblical Thomism is and should continue the dialogue with Ressourcement theology, Jewish theology, with ecumenical biblical theologies. You are right that just indicating three points someone could have an impression that there are only these three paths, while there are mucho more! I will drop them from the article in order to avoid misunderstandings.
Reviewer 3 Report
I would take "Finally, an example of a proposed approach (quies Dei) is analyzed," out of your abstract and out of your introduction, or expand that section and give it its own major heading. In your abstract and intro, it sounds like this will be a major section in your paper, but as it currently reads, it functions little more than as an example of, and elaboration on, your description of how biblical Thomism works.
I think the paper is a worthy contribution as it stands, and therefore just recommending you reword or take that bit out. It is ok that this paper is largely a "field consolidator" of sorts.
Author Response
Thank you for your comments: the third part - on quies Dei - can be somehow bigger, but my idea is to offer a short overview of the method of Biblical Thomism. Because I have published a detailed study on the method of BT, I will indicate this to readers.